# Unintended exposure to e-liquids and subsequent health outcomes among US youth and adults

**Juhan Lee**[1]*, **Grace Kong**[1], **Suchitra Krishnan-Sarin**[1], **Deepa R. Camenga**[2]

**1** Department of Psychiatry, Yale University School of Medicine, New Haven, CT, United States of America, **2** Department of Emergency Medicine, Yale University School of Medicine, New Haven, CT, United States of America

* Juhan.lee@yale.edu

## Abstract

This study aimed to determine the prevalence and predictors of oral, ocular, or dermal e-liquid exposure and subsequent outcomes (becoming sick, going to the hospital) in the US. We examined survey data from the Population Assessment of Tobacco and Health Study Wave 5 (2018–2019). The analytic sample included US youth (aged 12–17 years), young adults (aged 18–24 years), and older adults (aged ≥ 25 years) who reported e-cigarette use in the past 12 months. We first determined the prevalence of self-reported e-liquid exposure (in the mouth, skin, or eyes), subsequently "becoming sick" from the exposure, and "going to the hospital" after the exposure. We also examined associations between these outcomes and the device type used (refillable tank /mod system, replaceable prefilled cartridges, disposable/ other device type). E-liquid exposure was reported by 25% of youth (aged 12–17 years), 25% of young adults (aged 18–24 years), and 19% of older adults (aged≥ 25 years). Among individuals reporting e-liquid exposure, subsequent sickness was reported by 10% of youth11% of young adults, and 14% of older adults, and "going to the hospital" was reported by 3.5% of youth, 2.7% of young adults, and 6.8% of older adults. Among young adults, the use of a refillable tank /mod system was associated with higher odds of e-liquid exposure (aOR = 2.2, 95% CI = 1.2, 4.1) than the use of other device types, including disposables. The findings suggest that, at a minimum, e-cigarettes/e-liquids may need warning labels that state the risks of e-liquid exposure and packaging regulations that promote device and bottle designs that minimize e-liquid spills.

## Introduction

Between 2011 and 2024, 50,743 e-cigarette- and nicotine liquid-related exposure cases were reported to Poison Control Centers in the United States [1]. Gastrointestinal, ocular, and dermal exposure to Liquids in e-cigarettes (e-liquids) that contain nicotine, flavoring, and other chemicals [2] can contribute to acute nicotine poisoning and chemical burns in the skin and eyes [2]. To prevent these potential injuries, the U.S. Child Nicotine Poisoning Prevention Act,

**Data Availability Statement:** The PATH Study is publicly available at https://www.icpsr.umich.edu/web/NAHDAP/studies/36498.

**Funding:** The research reported in this publication was supported by grant number U54DA036151

from the National Institute on Drug Abuse (NIDA) and U.S. Food and Drug Administration (FDA) Center for Tobacco Products (CTP). Support for GK and JL is also provided by R01DA049878. There was no additional external funding received for this study. The funders had no role in study design, data collection and analysis, decision to publish, or preparation of the manuscript.

**Competing interests:** The authors have declared that no competing interests exist.

passed in 2015, requires manufacterers to use child-proof packaging for e-liquids [3]. Soon after the passage of this Act, the number of reported cases of e-liquid exposure among children aged 5 and below substantially decreased [4]. It is unknown how common e-liquid exposures are among older age groups and whether additional strategies are needed to prevent exposures in these populations.

This study aimed to determine the prevalence of self-reported e-liquid exposure (in the mouth, skin, or eyes), subsequently "becoming sick" from the exposure, and "going to the hospital" after the exposure among US youth (aged 12–17 years), young adults (aged 18–24 years), and older adults (aged ≥ 25 years) who used e-cigarettes. We chose to stratify the analyses by these age groups because previous research has shown that e-cigarette use behaviors (e.g. preferred device type, average frequency of use) vary by age [5, 6]. For example, adolescents (vs. adults) may be more likely to modify e-cigarette devices, [7] and e-cigarette use is prevalent among young adults (aged 18–24 years) [6]. These behaviors may put adolescents and young adults at risk for greater exposure to e-liquids. We also examined associations between the outcomes and the device type used (refillable tank /mod system, replaceable prefilled cartridges, disposable/other device type) to explore whether the ability to manually manipulate e-liquids through certain product features might affect the risk of exposure. Understanding e-liquid exposure and related outcomes by device types is important since individuals can access a variety of e-cigarette device types in the United States and e-cigarette popularity has been changed in recent years such that disposable and mod systems are frequently used by adolescents and young adults in recent years [8]. Further, mod systems might put people at greater risk for e-liquid exposure since people are directly putting e-liquids in their devices. Although closed devices such as disposables/cartridges have lower likelihood of e-liquid exposure since e-liquid is prefilled, research shows that youth and young adults are manipulating or 'hacking' these devices to put their own e-liquids, which may increase their e-liquid exposure [7].

## Materials and methods

We conducted a secondary data analysis of the US nationally representative Population Assessment of Tobacco and Health (PATH) Study Wave 5 (2018–2019) youth (aged 12–17 years) and adult (aged 18 years and older) datasets. The PATH Study public dataset's self-report measures are well-validated, and the details of the study methodology, eligibility criteria, measures, and theoretical framework are described elsewhere [9, 10]. The PATH uses multi-stage sampling to allow the calculation of US population estimates. Due to the unique sampling weights of each dataset, we were unable to combine the youth and adult datasets. This study was deemed exempt from the Yale Institutional Review Board.

### Measures

Past 12-month e-cigarette use was assessed with "In the past 12 months, have you used an electronic nicotine product, even one or two times? (Electronic nicotine products include e-cigarettes, vape pens, personal vaporizers and mods, e-cigars, e-pipes, e-hookahs, and hookah pens.)" with response options "yes" or "no."

E-liquid exposure was assessed among those who used e-cigarettes in the past 12 months with "In the past 12 months, did you swallow e-liquid or get it in your mouth, on your skin, or in your eyes?" [yes/no]. When a respondent reported "yes," they were subsequently asked, "In the past 12 months, after you swallowed e-liquid, or got it in your mouth, on your skin, or in your eyes, (1) did you become sick?" [yes/no], and (2) "did you go to a hospital?" [yes/no].

Past-30-day e-cigarette use was assessed with "In the past 30 days, on how many days did you use an electronic nicotine product?" [0–30 days]. Among individuals who reported using

e-cigarettes 1 or more days in the past 30 days, the "most often used device type" was assessed with the following response options: "a disposable device", "a device that uses replaceable pre-filled cartridges", "a device with a tank that you refill with liquids", "a mod system", "something else." Device types were collapsed into three groups due to small sample sizes: refillable tank system/mod system, replaceable prefilled cartridges, and disposable/other device type (set as reference group due to the hypothesis that there is less likelihood of exposure with closed-system device types).

### Analytical strategy

Among youth and adults who reported e-cigarette use in the past 12 months, we estimated the prevalence of e-liquid exposure, subsequently getting sick, and going to the hospital after the exposure. Among those who reported e-cigarette use in the past 30 days, we conducted separate multivariable binomial logistic regression models examining associations between each outcome (exposure, getting sick, going to the hospital) and "the most often used device type". Multivariable models were adjusted for sociodemographic covariates (i.e., sex, race, ethnicity, income level, education level, insurance status) and past 30-day e-cigarette use frequency. In the youth sample, education and income covariates refer to parental education and household annual income (reported by parents), and the insurance status was not recorded (Please see **Table 1**).

We used a complex sampling weight provided by PATH Study, which incorporates a multistage, stratified sampling design of PATH Study. For the variance estimation, the PATH Study datasets use a balanced repeated replication method of sampling weight, which selects subsamples repeatedly from the whole sample and calculates the statistics for each subsample, and then uses these replicate statistics of subsamples to estimate the variance of the full-sample statistic. We conducted hypothesis testing for 3 different outcomes. To adjust for multiple comparisons to prevent Type I error, we applied a Bonferroni correction and $p<0.017$ (0.05/3) was used as a threshhold for statistical significance [11]. We used STATA 18.0 (College Station, TX) for statistical analyses.

### Results

Among youth (ages 12–17) who used e-cigarettes in the past 12 months, 25.5% reported e-liquid exposure. Of those who reported exposure, 10.3% reported becoming sick, and 3.5% reported going to a hospital. Among young adults (ages 18–24) who used e-cigarettes in the past 12 months, 25.2% reported e-liquid exposure. Of those who reported exposure, 11.0% reported getting sick, and 2.7% reported they went to a hospital. Among older adults (ages ≥25) who used e-cigarettes in the past 12 months, 19.1% reported e-liquid exposure. Of those who reported exposure, 14.0% reported getting sick, and 6.8% reported that they went to a hospital (**Table 1**).

**Table 2** shows the results of associations between e-liquid exposure and subsequent outcomes and device type after adjusting for other covariates. Among young adults, the use of refillable tank/mod systems (vs. disposable/other device type) was associated with higher odds of e-liquid exposure (aOR = 2.2, 95% CI = 1.2, 4.1). Among older adults who were exposed to e-liquid, use of refillable tank/mod systems (aOR = 0.1, 95% CI = 0.04, 0.5) or replaceable pre-filled cartridge devices (aOR = 0.1, 95% CI = 0.02, 0.5) (vs. disposable/other device type) was associated with lower odds of becoming sick after an e-liquid exposure.

### Discussion

Using data from the nationally representative PATH study, we found that e-liquid exposure is common; up to 25% of individuals across all ages who used e-cigarettes in the past 12 months

**Table 1. Sample characteristics among past-12-month e-cigarette users.**

| Unweighted n (Weighted %) unless otherwise indicated | Youth (ages12-17) | Young adults (ages18-24) | Older adults (ages≥25) |
|---|---|---|---|
| **Sex** | | | |
| Female | 531 (49.7) | 710 (37.8) | 865 (44.4) |
| Male | 548 (50.3) | 1112 (62.1) | 881 (55.6) |
| **Race** | | | |
| White | 796 (80.6) | 1341 (79.5) | 1289 (77.0) |
| Black | 83 (6.5) | 169 (8.6) | 246 (13.8) |
| Others | 162 (13.0) | 250 (11.9) | 186 (9.2) |
| **Ethnicity** | | | |
| Non-Hispanic | 796 (80.9) | 1425 (84.1) | 1455 (85.9) |
| Hispanic | 246 (19.1) | 382 (15.9) | 265 (14.1) |
| **Household Annual Income** | | | |
| <$50,000 | 448 (40.5) | 1063 (61.4) | 1085 (63.5) |
| ≥$50,000 | 601 (59.5) | 614 (38.6) | 592 (36.5) |
| **Education** | | | |
| Completed college (no) | 679 (60.2)[a] | 991 (48.7) | 790 (49.3) |
| Completed college (yes) | 398 (39.8)[a] | 825 (51.3) | 946 (50.7) |
| **Has Health Insurance [b]** | | | |
| No | - | 451 (22.5) | 350 (20.3) |
| Yes | - | 1345 (77.5) | 1386 (79.7) |
| **Number of days of e-cigarette use in the past 30 days** | | | |
| 0–30 days, Mean(SD) | 10.6 (11.0) | 4.7 (7.9) | 3.3 (9.7) |
| **Most often used device type** | | | |
| Disposable/other device | 72 (6.8) | 104 (5.7) | 117 (7.7) |
| Refillable Tank /mod system | 430 (38.7) | 837 (50.4) | 955 (63.4) |
| Replaceable prefilled cartridges | 559 (54.4) | 716 (43.9) | 422 (28.9) |
| **Study Outcomes [c]** | | | |
| E-liquid Exposure (Yes) | 457 (25.5) | 999 (25.2) | 702 (19.1) |
| Sickness after Exposure (yes) [d] | 49 (10.3) | 114 (11.0) | 96 (14.0) |
| Went to hospital after Exposure (yes) [d] | 18 (3.5) | 29 (2.7) | 46 (6.8) |

a: In the youth sample, education and income covariates refer to parental education and household annual income (reported by parents); b: youth insurance status was not included in the youth dataset; c: assessed among past 12 month e-cigarette users; d: assessed among those with e-liquid exposure

report oral, ocular, or dermal e-liquid exposure. Among those who were exposed to e-liquids, 10% (youth), 11% (young adults), and 14% (older adults) reported becoming sick from the exposure, yet few reported going to the hospital after the exposure. Use of mod/tank systems was associated with e-liquid exposure in young adults, but not youth or older adults. Although we could not directly assess why individuals were exposed to e-liquids (e.g., accidental spills, device malfunction), these findings suggest that additional regulation of e-cigarette devices and e-liquids may be needed to prevent potential injuries, poisonings, and adverse health consequences [2].

We observed age-related differences prevalence rates of e-liquid exposures. For example, youth and young adults had a higher prevalence of e-liquid exposure than older adults. Further, higher e-cigarette use frequency was related to higher odds of e-liquid exposure, but this association was not observed in older age groups. Although increased e-cigarette use frequency may be related to frequent refills of e-liquids in both younger and older individauls, it is

**Table 2. Logistic regression models on e-liquid exposure and subsequent outcomes among past 30-day e-cigarette users.**

| | E-liquid Exposure | | | | | | Getting sick after the exposure [a] | | | | | | Going to a hospital after the exposure [a] | | | | | |
|---|---|---|---|---|---|---|---|---|---|---|---|---|---|---|---|---|---|---|
| | Youth | | Young adults | | Older adults | | Youth | | Young adults | | Older adults | | Youth | | Young adults | | Older adults | |
| | Adjusted Odds Ratio (95% CI) | p | Adjusted Odds Ratio (95% CI) | p | Adjusted Odds Ratio (95% CI) | p | Adjusted Odds Ratio (95% CI) | p | Adjusted Odds Ratio (95% CI) | p | Adjusted Odds Ratio (95% CI) | p | Adjusted Odds Ratio (95% CI) | p | Adjusted Odds Ratio (95% CI) | p | Adjusted Odds Ratio (95% CI) | p |
| **Sex (Ref = Female)** | | | | | | | | | | | | | | | | | | |
| Male | 0.8 (0.5, 1.2) | 0.37 | 0.9 (0.7, 1.2) | 0.71 | 1.1 (0.8, 1.5) | 0.40 | 0.7 (0.3, 1.8) | 0.52 | 1.2 (0.6, 2.1) | 0.52 | 1.2 (0.5, 2.7) | 0.60 | 1.0 (0.2, 5.1) | 0.92 | 2.1 (0.3, 11.8) | 0.38 | 3.2 (0.8, 12.1) | 0.08 |
| **Race (Ref = White)** | | | | | | | | | | | | | | | | | | |
| Black | 0.6 (0.2, 1.4) | 0.26 | **0.6 (0.4, 0.9)** | 0.04 | 0.7 (0.4, 1.2) | 0.20 | 3.5 (0.9, 13.4) | 0.06 | 1.1 (0.4, 2.8) | 0.81 | 1.6 (0.5, 4.7) | 0.35 | 4.3 (0.3, 7.8) | 0.22 | 2.4 (0.4, 12.5) | 0.29 | 1.8 (0.2, 11.4) | 0.52 |
| Others | 0.8 (0.5, 1.4) | 0.58 | 1.1 (0.7, 1.6) | 0.51 | 1.2 (0.7, 1.9) | 0.44 | 2.3 (0.6, 8.9) | 0.20 | 0.6 (0.2, 1.7) | 0.42 | 1.2 (0.1, 0.6) | 0.83 | 0.9 (0.1, 5.6) | 0.92 | 0.2 (<0.01, 2.8) | 0.26 | 4.4 (0.4, 48.1) | 0.21 |
| **Ethnicity** (Ref = Non-Hispanic) | | | | | | | | | | | | | | | | | | |
| Hispanic | 1.3 (0.8, 2.0) | 0.20 | **0.6 (0.4, 0.8)** | <.01 | 0.6 (0.4, 1.0) | 0.07 | 1.3 (0.4, 3.8) | 0.60 | 0.9 (0.3, 2.5) | 0.93 | 1.4 (0.3, 5.9) | 0.62 | 2.5 (0.5, 1.1) | 0.22 | 1.6 (0.2, 10.7) | 0.61 | 1.9 (0.2, 15.6) | 0.52 |
| **Household Annual Income** (Ref = <$50,000) | | | | | | | | | | | | | | | | | | |
| ≥$50,000 | 1.0 (0.7, 1.4) | 0.95 | **1.6 (1.1, 2.1)** | <.01 | 1.2 (0.8, 1.8) | 0.27 | 0.4 (0.1, 1.2) | 0.14 | 0.8 (0.4, 1.6) | 0.71 | 0.9 (0.3, 2.2) | 0.90 | 0.1 (0.01, 1.7) | 0.11 | 0.1 (0.003, 0.8) | 0.03 | 0.7 (0.2, 2.3) | 0.62 |
| **Education** [b] (Ref = <College) | | | | | | | | | | | | | | | | | | |
| ≥ College | 1.5 (1.0, 2.3) | 0.03 | 1.1 (0.8, 1.5) | 0.42 | 0.9 (0.7, 1.3) | 0.87 | 0.9 (0.3, 2.2) | 0.84 | 0.7 (0.3, 1.3) | 0.27 | 0.4 (0.1, 1.0) | 0.05 | 0.5 (0.03, 0.4) | 0.70 | 0.4 (0.05, 2.9) | 0.36 | 0.2 (0.05, 1.0) | 0.05 |
| **Having an insurance** [c] (Ref = No) | | | | | | | | | | | | | | | | | | |
| Yes | - | - | 1.2 (0.9, 1.7) | 0.12 | 1.0 (0.7, 1.5) | 0.59 | - | - | 0.6 (0.3, 1.3) | 0.26 | 1.2 (0.4, 3.2) | 0.65 | - | | 0.5 (0.1, 1.9) | 0.35 | 0.4 (0.1, 1.9) | 0.29 |
| **Past 30-day e-cigarette use frequency** | | | | | | | | | | | | | | | | | | |
| 0–30 | **1.1 (1.0, 1.1)** | <0.01 | 0.9 (0.9, 1.0) | 0.07 | 0.9 (0.9, 1.0) | 0.38 | 1.0 (0.9, 1.0) | 0.60 | 1.0 (0.9, 1.0) | 0.73 | 0.9 (0.9, 1.0) | 0.49 | 1.0 (0.9, 1.1) | 0.18 | 1.0 (0.9, 1.0) | 0.92 | 1.0 (0.9, 1.0) | 0.98 |
| **Most often used device type** (ref = Disposable/other device) | | | | | | | | | | | | | | | | | | |
| Refillable Tank /mod system | 2.1 (0.9, 4.9) | 0.07 | **2.2 (1.2, 4.1)** | <.01 | 2.1 (1.1, 4.2) | 0.02 | 0.8 (0.05, 14.4) | 0.89 | 0.2 (0.09, 0.8) | 0.02 | **0.1 (0.04, 0.5)** | <.01 | 1.1 (0.1, 14.8) | 0.88 | 0.2 (0.04, 1.6) | 0.15 | 0.1 (0.01, 0.7) | 0.02 |
| Replaceable prefilled cartridges | 1.6 (0.7, 3.7) | 0.22 | 2.0 (1.1, 3.6) | 0.02 | 1.4 (0.7, 3.0) | 0.26 | 0.9 (0.05, 17.8) | 0.96 | 0.5 (0.1, 1.4) | 0.20 | **0.1 (0.02, 0.5)** | <.01 | 0.5 (0.04, 7.4) | 0.67 | 0.1 (0.02, 1.3) | 0.09 | 0.0 (0.004, 0.7) | 0.02 |

Youth (ages 12–17), Young adults (ages 18–24), Older adults (ages≥25)

aOR (95% CI), p-value indicated; all significant findings had Lower Confidence Interval >0

**Note:** Most often used device type was only assessed among current (past-30-day) e-cigarette users.

A statistical cut-off for p-value was set as 0.017 (0.05/3 = 0.017) for multiple comparisons for 3 outcomes

a: questions are only asked to those who were exposed to e-liquid; b: parental education question was asked to parents in the youth dataset; c: insurance status was lacking in the youth dataset.

possible that younger individuals are less experienced than older adults in safely handling e-cigarette devices and liquids,. It is worth noting that although older adults had the lowest prevalence of e-liquid exposure tthey reported the highest prevalence of "becoming sick" and "going to a hospital" after the exposure This may suggest that the severity of e-liquid exposure might be worse among older adults than younger people, necessitating a greater level of medical attention. Alternatively, youth may experience similar types of e-liquid exposure, but be less likely to present to the hostipal due to confidentiality concerns to the hospitals and may hide their e-liquid exposure from parents and may go to the hospital for greater symptom severity. Future studies should use clinical data (e.g., electronic health records) to identify specific symptoms and severity of symptoms related to e-liquid exposure.

We also found that the use of refillable tank/mod systems or replaceable prefilled cartridges (vs. disposable / other device type) was associated with higher odds of e-liquid exposure among young adults. It is plausible that e-liquid exposures with these specific devices may have arisen in the context of mixing e-liquids, changing cartridges, or modifying e-cigarette devices [12]. Interestingly, use of refillable tank/mod systems or replaceable prefilled cartridges (vs. disposable / other device type) was also associated with lower odds of becoming sick from e-liquid exposures among older adults. This finding suggests that older adults may refill or replace their e-liquids using potentially safer practices than young adults; future studies are needed to understand which behaviors increase the risk for exposure among both these populations.

Our study also highlights sociodemographic disparities in e-liquid exposures. Among young adults, Non-Hispanic individuals and those with higher income levels had higher odds of e-liquid exposure to e-liquid. In general, more frequent use of e-cigarettes may be associated with increased risks of e-liquid exposure. Existing data indicate that e-cigarettes are more likely to be used by non-Hispanic White individuals, and those from higher income levels among young adults [6]. Thus, certain education and communication related to harmful effects and risks of e-liquid exposure could be targeted to populations who are at greater risk for using e-cigarettes.

## Limitations

This study has several limitations. PATH survey data may be prone to self-report, recall, or social desirability bias. Survey questions did not assess the specific route of exposure, the severity of the exposure or subsequent sickness, or if going to the hospital resulted in hospital admission or other medical interventions. Future studies should examine these factors since the health impact and harm could vary by route and extent of exposure. This consideration is important since the health impact and harm could vary by route and extent of exposure. Additionally, these data were collected in 2018–2019 and may not reflect current e-cigarette-related behaviors among US youth and adults. Findings may not generalize to people who use multiple device types, previous e-cigarette users (we only included individuals who used e-cigarettes in the past 12 months) or individuals outside of the U.S. Given the broad reach of the use of e-cigarettes, future research is warranted with data from other countries.

## Conclusions

This study provides one of the first estimates of e-liquid exposure and potential health burdens from the exposure among youth and adults using US nationally representative samples. Overall, the findings highlight the need for effective e-liquid packaging and labeling regulations. This could include warning labels to inform potential risk of e-liquid exposure, e-cigarette device, and e-liquid bottle design to prevent accidental exposure to e-liquids. Our findings also

suggest that e-liquid injury prevention strategies may need to be directed toward refillable tank/mod systems and individuals who frequently use e-cigarettes. Moreoever, future research is needed to better characterize the clinical severity of e-liquid exposures and how they occur (e.g., while refilling or mixing e-liquids).

## Author Contributions

**Conceptualization:** Juhan Lee.

**Formal analysis:** Juhan Lee.

**Funding acquisition:** Grace Kong, Suchitra Krishnan-Sarin.

**Investigation:** Juhan Lee, Grace Kong, Suchitra Krishnan-Sarin, Deepa R. Camenga.

**Supervision:** Deepa R. Camenga.

**Writing – original draft:** Juhan Lee.

**Writing – review & editing:** Juhan Lee, Grace Kong, Suchitra Krishnan-Sarin, Deepa R. Camenga.

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
