## [Decision Letter · Decision Letter 0]

12 Sep 2023

PONE-D-23-26133Unintended exposure to e-liquid and subsequent health outcomes among US youth and adultsPLOS ONE

Dear Dr.  Lee,

Thank you for submitting your manuscript to PLOS ONE. After careful consideration, we feel that it has merit but does not fully meet PLOS ONE’s publication criteria as it currently stands. Therefore, we invite you to submit a revised version of the manuscript that addresses the points raised during the review process.

ACADEMIC EDITOR:Need extensive revision on method section and need to add background information on result section. Older Adults: What is the Maximum age?Some 95% CI have a 0 lower limit. Can you please explain, why?The authors have not presented the demographic characteristics of the study population which is very crucial to understand and reach a conclusion on logistic regression. Authors are required to present the Detail analysis plan in the Method section. How complex analysis was performed? How did you define sampling weight? Why authors not present univariate analysis prior to applying logistic regression?Please submit your revised manuscript by 5th October 2023. If you will need more time than this to complete your revisions, please reply to this message or contact the journal office at plosone@plos.org. Please include the following items when submitting your revised manuscript:A rebuttal letter that responds to each point raised by the academic editor and reviewer(s). You should upload this letter as a separate file labeled 'Response to Reviewers'.A marked-up copy of your manuscript that highlights changes made to the original version. You should upload this as a separate file labeled 'Revised Manuscript with Track Changes'.An unmarked version of your revised paper without tracked changes. You should upload this as a separate file labeled 'Manuscript'.

We look forward to receiving your revised manuscript.

Kind regards,

Umesh Raj Aryal, PhD

Academic Editor

PLOS ONE

Journal Requirements:

2.Thank you for stating in your Funding Statement: 

The research reported in this publication was supported by grant number U54DA036151 from the National Institute on Drug Abuse (NIDA) and U.S. Food and Drug Administration (FDA) Center for Tobacco Products (CTP). Support for GK and JL is also provided by R01DA049878. 

**Comments to the Author**

1. Is the manuscript technically sound, and do the data support the conclusions?

Reviewer #1: Yes

Reviewer #2: Yes

2. Has the statistical analysis been performed appropriately and rigorously? 

Reviewer #1: Yes

Reviewer #2: Yes

3. Have the authors made all data underlying the findings in their manuscript fully available?

Reviewer #1: Yes

Reviewer #2: Yes

4. Is the manuscript presented in an intelligible fashion and written in standard English?

Reviewer #1: Yes

Reviewer #2: Yes

5. Review Comments to the Author

Reviewer #1: In results, obviously a key area, it should be made very clear that you present initial percentage for reported exposure. The next sentence could have a phrase like "Of those who reported exposure, ....". This would make it very clear to a speed reader that 25% or so of subjects reported exposure and that of these, 11% or xx% of all subjects reported becoming sick"

It is worth mentioning somewhere that the analysis was of recent users and that an uncertain number of previous users may have had an exposure that triggered a cessation attempt. They will not be in the numerator or denominator for this analysis

You are right to point out that devices have changed with the rise of the cheap disposable high nicotine salt concentration products. Unless the device is taken apart, exposures are unlikely from these devices - still a risk for young children who pick up and suck but this risk is not assessed in this paper

Reviewer #2: I thank the authors for the opportunity to review this manuscript; it's very well written and straight to the point. I had a few comments/suggestions about the methods for your consideration.

Methods: Can the authors clarify exactly what the analytic sample is and provide a clear justification in the methods; in the methods they mention past 12 month e-cigarette use and the variable past-30-day e-cigarette use. Was past month e-cigarette use measured a continuous variable? I ask this because the table presents this as a frequency variable; however the methods do not tell the reader how this variable was measured.

The authors mention e-cigarette related factors were controlled for? Besides past 30 -day e-cigarette use, the other variables appear to be sociodemographic factors. The way it is written would lead the reader to "expect" a couple of e-cigarette related factors.

I also think that some of the information presented on the footnote of Table 1; should be mentioned in the methods of the paper. The methods should be better explained for reproducibility.

Results: The authors refer us to Table 1 for the percentages and this was not presented in Table 1.

6. PLOS authors have the option to publish the peer review history of their article (what does this mean?). If published, this will include your full peer review and any attached files.

Reviewer #1: No

Reviewer #2: No

---

## [Author Response · Author response to Decision Letter 0]

13 Nov 2023

Manuscript ID: PONE-D-23-26133

Title: Unintended exposure to e-liquid and subsequent health outcomes among US youth and adults

Editor

1. Need extensive revision on the method section and need to add background information on the result section.

Response: 

Thank you for your comment. We revised our method section and added more background information in the results section. Please see responses to the Reviewers’ comments below for details.

Revised text: 

Page 4-7, Please see revised Methods and Results section.

2. Older Adults: What is the Maximum age?

Response: 

The age variable was a recoded categorical variable from the PATH Study Public Use dataset [ “1 = 18 to 24 years old, 2 = 25 to 34 years old, 3 = 35 to 44 years old, 4 = 45 to 54 years old, 5 = 55 to 64 years old, 6 = 65 or more years old.”]. The recoded variable did not include an upper age limit so we are not able to report on this data.

3. Some 95% CI have a 0 lower limit. Can you please explain why?

Response: 

The lower confidence of population estimate for several odd ratios was presented as 0 due to small cell sizes. However, those lower confidence intervals are actually not exactly zero. For instance, one of the confidence intervals was 0.00464, which was rounded up to 0. All significant findings had lower confidence interval>0. To reduce the confusion, we revised the Table (i.e., indicated <0.01 for the lower CI). 

Revised text: Please see updated Table.

4. The authors have not presented the demographic characteristics of the study population which is very crucial to understand and reach a conclusion on logistic regression. Authors are required to present the Detail analysis plan in the Method section. How complex analysis was performed? How did you define sampling weight? Why did the authors not present univariate analysis prior to applying logistic regression?

Response: 

Thank you for your questions. We added information about the detailed analysis plan (e.g., how the analysis was conducted, complex sampling weight) and presented the results of univariate analyses and sample characteristics in Table 2.

Revised text:

Page 4-7, Please see revised Methods and Results section.

Reviewer #1: 

1. In results, obviously a key area, it should be made very clear that you present initial percentage for reported exposure. The next sentence could have a phrase like "Of those who reported exposure, ....". This would make it very clear to a speed reader that 25% or so of subjects reported exposure and that of these, 11% or xx% of all subjects reported becoming sick."

Response: 

Thank you for your suggestions. We rewrote the result section per the reviewer’s recommendations.

Revised text:

Page 6, “Among youth who used e-cigarettes in the past 12 months, 25.5% reported e-liquid exposure during this time. Of those who reported exposure, 10.3% reported becoming sick from the exposure, and 3.5% reported going to a hospital after the exposure. Among young adults (ages 18-24) who used e-cigarettes in the past 12 months, 25.2% reported e-liquid exposure during this time. Of those who reported exposure, 11.0% reported getting sick, and 2.7% reported they went to a hospital. Among older adults (ages ≥25) who used e-cigarettes in the past 12 months, 19.1% reported e-liquid exposure during this time. Of those who reported exposure, 14.0% reported getting sick, and 6.8% reported that they went to a hospital (Table 1).”

2. It is worth mentioning somewhere that the analysis was of recent users and that an uncertain number of previous users may have had an exposure that triggered a cessation attempt. They will not be in the numerator or denominator for this analysis

Response: 

Thank you for your suggestion. We added this to the limitation section.

Revised text:

Page 8, “This study does not capture e-liquid exposures among previous e-cigarette users as we only included individuals who used e-cigarettes in the past 12 months. It is possible that individuals may have stopped using e-cigarettes due to being sick from exposure.”

3. You are right to point out that devices have changed with the rise of the cheap disposable high-nicotine salt concentration products. Unless the device is taken apart, exposures are unlikely from these devices - still a risk for young children who pick up and suck but this risk is not assessed in this paper

Response: 

Thank you for bringing this up. We agree that disposable devices still can pose a risk to young children, as they are developmentally likely to put objects in their mouths. However, we think this is out of the scope of this study since we only assessed the risk of e-liquid exposure among youth and adults.

Reviewer #2: 

1. Methods: Can the authors clarify exactly what the analytic sample is and provide a clear justification in the methods; in the methods they mention past 12 month e-cigarette use and the variable past-30-day e-cigarette use. 

Response: 

We added a more detailed description of our analytic sample to the Methods. In brief, the analytic sample included past-12-month e-cigarette users because the outcome variables were measured in this time frame (“In the past 12 months, did you swallow e-liquid or get it in your mouth, on your skin, or in your eyes?”). However, we only included past 30-day e-cigarette users in the device type analyses because the device type question was only measured in this group.

Revised text:

Page 5, “We selected past-30-day e-cigarette users as an analytic sample in these regression models since the most often used device type was only asked to past-30-day e-cigarette users.”

2. Was past month e-cigarette use measured a continuous variable? I ask this because the table presents this as a frequency variable; however the methods do not tell the reader how this variable was measured.

Response: 

“Past month e-cigarette use” was measured as a continuous variable. We indicated this in the text of revision.

Revised text:

Page 5, “The past-30-day e-cigarette use was assessed as “In the past 30 days, on how many days did you use an electronic nicotine product?” This was a continuous variable and ranged from 0 to 30.”

3. The authors mention e-cigarette related factors were controlled for. Besides past 30 -day e-cigarette use, the other variables appear to be sociodemographic factors. The way it is written would lead the reader to "expect" a couple of e-cigarette related factors.

Response: We revised this sentence.

Revised text:

Page 5, “We selected past-30-day e-cigarette users as an analytic sample in these regression models since the most often used device type was only asked to past-30-day e-cigarette users. The covariates included sociodemographics (i.e., sex, race, ethnicity, income level, education level, having insurance) and past-30-day e-cigarette use frequency (See Table 1).”

4. I also think that some of the information presented on the footnote of Table 1; should be mentioned in the methods of the paper. The methods should be better explained for reproducibility.

Response: We added the information from the footnotes to the Methods

Revised text:

Page 5, “We selected past-30-day e-cigarette users as an analytic sample in these regression models since the most often used device type was only asked to past-30-day e-cigarette users.”

Page 5-6, “The covariates included sociodemographics (i.e., sex, race, ethnicity, income level, education level, insurance status) and past-30-day e-cigarette use frequency. For youth sample, parental education and parental income were used and the insurance status was lacking in the youth dataset (Please see Table 1).”

5. Results: The authors refer us to Table 1 for the percentages and this was not presented in Table 1.

Response: 

We revised the tables and text accordingly. Please note that Table 1 now presents the estimates for outcomes among past-12-month e-cigarette users, Table 2 now presents the results of bivariate analyses between outcomes and predictors, and Table 3 presents the results of multivariable regression models.

Revised text:

Page 6-7, Please see updated Result section.

---

## [Editor Report · Decision Letter 1]

24 Nov 2023

PONE-D-23-26133R1Unintended exposure to e-liquid and subsequent health outcomes among US youth and adultsPLOS ONE

Dear Dr. Lee,,

Thank you for submitting your revised version manuscript to PLOS ONE. After careful consideration, we feel that it has merit but does not fully meet PLOS ONE’s publication criteria as it currently stands. Therefore, we invite you to submit a revised version of the manuscript that addresses the points raised during the review process.

Comments:

Line 105-108 discussed demographic variables but Table 1 does not have all this information. Please update Table 1.

Table 2 is still unclear. Authors have presented % but p values and CI are missing to confirm significance. If the table is going to big, the Authors can provide details in the Supplementary table. 

 "A statistical cut-off for p-value was set as 0.017 (0.05/3=0.017) for multiple comparisons for 3 outcomes". Please Justify it. 

We look forward to receiving your revised manuscript.

Kind regards,

Umesh Raj Aryal, PhD

Academic Editor

PLOS ONE

Journal Requirements:

Additional Editor Comments:

Line 105-108 discussed demographic variables but Table 1 does not have all this information. Please update Table 1.

Table 2 is still unclear. Authors have presented % but p values and CI are missing to confirm significance. If the table is going to be big, the Authors can provide details in the Supplementary table.

"A statistical cut-off for p-value was set as 0.017 (0.05/3=0.017) for multiple comparisons for 3 outcomes". Please Justify it.

---

## [Author Response · Author response to Decision Letter 1]

14 Jan 2024

Manuscript ID: PONE-D-23-26133R1

Title: Unintended exposure to e-liquid and subsequent health outcomes among US youth and adults

1. Line 105-108 discussed demographic variables but Table 1 does not have all this information. Please update Table 1.

Response: We updated Table 1. Please see p. 9. 

2. Table 2 is still unclear. Authors have presented % but p values and CI are missing to confirm significance. If the table is too big, the Authors can provide details in the Supplementary table. 

Response: We added p-values in Table 2. Please see p. 10.

3. "A statistical cut-off for p-value was set as 0.017 (0.05/3=0.017) for multiple comparisons for 3 outcomes". Please Justify it.

Response: We added justification for multiple comparison adjustments. 

Revised text: Page 7, “We conducted hypothesis testing for 3 different outcomes. To adjust for multiple comparisons to prevent Type I error, we applied a Bonferroni correction and p<0.017 (0.05/3) was used as a threshhold for statistical significance).(Bland and Altman, 1995)”

---

## [Decision Letter · Decision Letter 2]

4 Apr 2024

PONE-D-23-26133R2Unintended exposure to e-liquid and subsequent health outcomes among US youth and adultsPLOS ONE

Dear Dr. Lee,

Thank you for submitting your manuscript to PLOS ONE. After careful consideration, we feel that it has merit but does not fully meet PLOS ONE’s publication criteria as it currently stands. Therefore, we invite you to submit a revised version of the manuscript that addresses the points raised during the review process.

We look forward to receiving your revised manuscript.

Kind regards,

Umesh Raj Aryal, PhD

Academic Editor

PLOS ONE

Journal Requirements:

Additional Editor Comments:

The reviewers have also provided us with constructive feedback and suggestions on methodology and other areas. We want to let you know that authors should respond to all of the feedback and comments.

Reviewers' comments:

Reviewer's Responses to Questions

**Comments to the Author**

1. If the authors have adequately addressed your comments raised in a previous round of review and you feel that this manuscript is now acceptable for publication, you may indicate that here to bypass the “Comments to the Author” section, enter your conflict of interest statement in the “Confidential to Editor” section, and submit your "Accept" recommendation.

Reviewer #3: (No Response)

Reviewer #4: (No Response)

Reviewer #5: All comments have been addressed

2. Is the manuscript technically sound, and do the data support the conclusions?

Reviewer #3: Partly

Reviewer #4: Yes

Reviewer #5: Partly

3. Has the statistical analysis been performed appropriately and rigorously? 

Reviewer #3: I Don't Know

Reviewer #4: Yes

Reviewer #5: No

4. Have the authors made all data underlying the findings in their manuscript fully available?

Reviewer #3: Yes

Reviewer #4: Yes

Reviewer #5: Yes

5. Is the manuscript presented in an intelligible fashion and written in standard English?

Reviewer #3: Yes

Reviewer #4: Yes

Reviewer #5: Yes

6. Review Comments to the Author

Reviewer #3: OVERALL COMMENTS

Note: I have commented on the second version of the manuscript (Revised Manuscript with Track Changes) in the file titled ‘Revision 2’.

This is a brief manuscript which addresses a clear question and provides a strong rationale. It uses publicly available data from a well-established large-scale nationally representative survey.

More context could be provided in the introduction. Rationale about why these specific age groups were compared would strengthen this paper, as would the use of more high-quality citations in the introduction and discussion. The use of consistent terminology (e.g., variable descriptors, device types) throughout would improve readability.

It would have been nice to see these analyses pre-registered (if they were, please include a reference to the pre-registration in Methods). The methods section is reasonably well structured although currently missing some key details. Information presented in the methods section does not need to be reiterated in the introduction or results.

To the extent of my knowledge, the analyses appear reasonably rigorous. The results are clearly presented, with three useful tables. Some repetition could be avoided between tables and text (particularly in the first paragraph of the results section).

Some seemingly important findings are not discussed. Frequency of use appears to be an important variable but other than saying it was a covariate I do not see it mentioned in the results or the discussion. Similarly, going to hospital rates are twice as high among older adults than the other age groups, but this is not discussed. Conclusions could be refined, please refer to the specific comments below.

SPECIFIC COMMENTS

ABSTRACT

I suggest including clear and specific definitions of groups and variables in the abstract, e.g., specifying the age range of ‘youth’, ‘young adults’, and ‘older adults’, defining ‘becoming sick’. The phrase “vs a disposable device and something else” needs to be clarified; I understand this is the reference group including two collapsed options but the current phrasing is confusing without context from methods. For instance, it could be phrased as: “vs people who used other device types including disposables” and combined in the same parentheses as the aOR/CI. Remove ‘however’ on the 9th line.

In the abstract the conclusion is a little narrow (warning labels only); consider other packaging regulations too, and even device/bottle designs which could avoid leaks (this can be explored in more details in the discussion, with the caveat that you don't know the exact cause of exposure).

INTRODUCTION

- First sentence: I cannot find the number of cases cited in the linked source. It would also help to specify the timeframe this number refers to (e.g., 3,864 new cases over the past year?)

- ‘children less than age 5’: Consider rephrasing (‘aged 5 and below’, ‘under the age of 5’…)

- Second to last sentence: ‘influence of device type used and exposure to e-liquids and on subsequent outcomes’.

- I suggest not specifying past 12 months/ past 30 days in this paragraph, because the discrepancy between to two raises questions which are not answered until the methods section, and it is redundant if specified below.

- Last sentence: Clearly state the comparators here; ‘tank and mod devices allow individuals to easily refill the device’ as compared to which devices? Disposables, all other devices? Providing some background on the types of devices available and their characteristics may strengthen the rationale for this hypothesis while also providing clarifications for those less familiar with the e-cig market.

METHODS

- Paragraph 1: Please provide additional details about PATH methodology or add a reference to a paper which does. All data used in this study was self-reported via surveys this should be specified in the opening paragraph.

- Please specify which statistical software/language was used

- “We selected past-30-day e-cigarette users as an analytic sample in these regression models since The most often used device type question was only asked to past-30-day e-cigarette users”

- “For the youth sample, education and income covariates refer to parental education and parental income, were used and the insurance status was not recorded. lacking in the youth dataset”

- “We used a complex sampling weight provided by PATH Study, which incorporates a multistage, stratified sampling design of PATH Study.”

- Last paragraph: which 3 outcomes?

RESULTS

- Paragraph 1 specified the age range for young adults and older adults but not youth; be consistent; no need to specify the ranges again here when you do so in methods (and abstract)

- Consider specifying the ‘covariates’ and ‘outcomes’ in Table 2’s title

- Paragraph 2: extra semi-colon before ‘in comparison’

- Paragraph 3: inconsistent variable descriptions: “(vs. disposable devices/ something else)” and “(vs. disposable/other devices)”

DISCUSSION

Paragraph 1:

- 7th line typo “e-liqiud”

- In the introduction it states “We hypothesized that using refillable tank or mod system e-cigarette devices would be associated with a higher likelihood of e-liquid exposures, given that these devices allow individuals to easily refill the device with e-liquids”, then in the discussion “Consistent with our hypothesis, use of refillable tank/mod systems or replaceable prefilled cartridges (vs. disposable devices or something else) was associated with higher odds of e-liquid exposure among young adults”. I don’t see how prefilled cartridge models are ‘consistent’ in this case; they were not mentioned in the hypothesis, and they do not require directly handling e-liquids.

- ‘becoming sick with devices with…’: rephrase e.g., ‘Older adults who used devices with refillable tanks, mod systems or replaceable prefilled cartridges had lower odds of becoming sick’

Paragraph 2:

- The limitations should mention how broad ‘becoming sick’ is as a measure

- “Did not assess .. whether it required hospital admission” – slightly confusing phrasing since it did assess if people went to hospital. Perhaps rephrase to clarify it didn’t assess if these hospitalisations were necessary?

- Could cite https://pubmed.ncbi.nlm.nih.gov/35622007/

- “especially directed at specific device types”: would be good to elaborate on this point

- Reference (9) needs context, not entirely clear why cited here

- “further research on how these exposures occur”: this is vague, I would opt for more specific phrasing, particularly as a closing statement

Reviewer #4: This analysis will be a valuable contribution to the literature; a few small suggestions are provided below.

Abstract. The final sentence suggests that warning labels informing users of the risk of e-cigarette liquid exposure are needed. While perhaps true, this would seem the very minimum needed to address the problem, which your data show is encountered by 20-25% of past-year users. Consider saying, "The findings suggest that, at a minimum, e-cigarettes/e-liquids..." Alternatively, consider a broader sentence similar to that which concludes the manuscript: "The findings also highlight the need for consistent and strong enforcement of e-liquid packaging and labeling regulations..."

Discussion:

Page 6, Re: "up to 25% of individuals...report oral, ocular, and dermal e-liquid exposure." Based on the PATH question, the "and" should be "or."

Page 6/7 Re: the phrase, ".... and few reported going to the hospital." I realize the numbers are small, but the phrasing understates your finding. Suggest either, "...a few reported..." OR "a small percentage of these reported..."

Finally, I would reiterate in the Discussion section that these findings are from a US based nationally representative sample. Given the broad reach of use of e-cigarettes, data from other countries will be useful.

Reviewer #5: This study provides valuable insights into the prevalence and potential consequences of e-liquid exposure. The followings are my suggestions for the authors.

1. The methods section lacks sufficient detail. Inclusion and exclusion criteria for the study population should be clearly outlined. Additionally, there is a lack of specificity regarding e-liquid exposure, including assessing the severity or extent of exposure, which could impact the resulting health consequences.

2. The study heavily relies on self-reported data from survey respondents, which introduces the possibility of recall bias or social desirability bias. This may lead to inaccurate reporting of e-liquid exposure and related outcomes, thereby affecting the reliability of the findings.

3. Potential confounding factors, such as nicotine dependence, risk-taking behaviors, or specific product characteristics, should have been addressed. Failure to account for these factors could compromise the validity of the study's conclusions regarding the prevalence and consequences of e-liquid exposure.

4. The mention of "Consistent with our hypothesis..." in line 121 implies the existence of a study hypothesis, which should have been clearly stated in the Introduction section for transparency and clarity.

5. Table 1 reports income data for youth aged 12-17, which raises questions about the validity of the survey instrument. Additionally, the mention of income for this age group seems out of context. If included, it should be clarified whether this refers to personal income or household income, and if it is indeed yearly income.

6. The implications of the study findings for tobacco control efforts should be thoroughly discussed. Specifically, how the results can contribute to better understanding and addressing e-liquid exposure among different age groups, and how this understanding can inform more effective tobacco control strategies.

In conclusion, due to the methodological shortcomings, it is recommended that this study be rejected.

7. PLOS authors have the option to publish the peer review history of their article (what does this mean?). If published, this will include your full peer review and any attached files.

Reviewer #3: No

Reviewer #4: No

Reviewer #5: No

---

## [Author Response · Author response to Decision Letter 2]

13 Jun 2024

PONE-D-23-26133R2: Unintended exposure to e-liquid and subsequent health outcomes among US youth and adults

Reviewer #3: 

1. This is a brief manuscript that addresses a clear question and provides a strong rationale. It uses publicly available data from a well-established, large-scale, nationally representative survey. More context could be provided in the introduction. Rationale about why these specific age groups were compared would strengthen this paper, as would the use of more high-quality citations in the introduction and discussion. 

Response: Thank you for your support of this paper. As suggested, we have added a rationale in the Introduction to include the three age strata within the analysis. 

We also include reputable citations from sources such as Centers for Disease Control and Prevention and the National Center for Chronic Disease Prevention and Health Promotion (US) Office on Smoking and Health.

See references:

Kramarow EA, Elgaddal N. Current Electronic Cigarette Use Among Adults Aged 18 and Over: United States, 2021. NCHS Data Brief. 2023;(475):1-8.

National Center for Chronic Disease Prevention and Health Promotion (US) Office on Smoking and Health. E-Cigarette Use Among Youth and Young Adults: A Report of the Surgeon General [Internet]. Atlanta (GA): Centers for Disease Control and Prevention (US); 2016. Available from: https://www.ncbi.nlm.nih.gov/books/NBK538680/

Revised text: 

Page 3, We chose to stratify the analyses by these age groups because previous research has shown that e-cigarette use behaviors (e.g. preferred device type, average frequency of use) vary by age.1,2 For example, adolescents (vs. adults) may be more likely to modify e-cigarette devices,3 and e-cigarette use is prevalent among young adults (aged 18-24 years).2 These behaviors may put adolescents and young adults at risk for greater exposure to e-liquids.

2. The use of consistent terminology (e.g., variable descriptors, device types) throughout would improve readability.

Response: We agree with this suggestion and have revised the text accordingly.

3. It would have been nice to see these analyses pre-registered (if they were, please include a reference to the pre-registration in Methods). 

Response: Unfortunately, this article has not been pre-registered. Plos One does not require pre-registration of secondary data analysis of publicly available national datasets.

4. The methods section is reasonably well structured, although currently missing some key details. Information presented in the methods section does not need to be reiterated in the introduction or results.

Response: We revised the Introduction, Methods and Results accordingly to reduce redundancies. 

Revised text: Please see updated Introduction and Results.

5. To the extent of my knowledge, the analyses appear reasonably rigorous. The results are clearly presented, with three useful tables. Some repetition could be avoided between tables and text (particularly in the first paragraph of the results section).

Response: We deleted many redundancies throughout the paragraph (e.g., “who used e-cigarettes in the past 12 months”).

Revised text:

Page 6, Among youth (ages 12-17) who used e-cigarettes in the past 12 months, 25.5% reported e-liquid exposure. Of those who reported exposure, 10.3% reported becoming sick, and 3.5% reported going to a hospital. Among young adults (ages 18-24) who used e-cigarettes in the past 12 months, 25.2% reported e-liquid exposure. Of those who reported exposure, 11.0% reported getting sick, and 2.7% reported they went to a hospital. Among older adults (ages ≥25) who used e-cigarettes in the past 12 months, 19.1% reported e-liquid exposure. Of those who reported exposure, 14.0% reported getting sick, and 6.8% reported that they went to a hospital (Table 1).

6. Some seemingly important findings are not discussed. Frequency of use appears to be an important variable, but other than saying it was a covariate, I do not see it mentioned in the results of the discussion. Similarly, going to hospital rates are twice as high among older adults than the other age groups, but this is not discussed. 

Response: Thank you for your suggestions. We added discussion about these findings.

Revised text:

Page 8, We observed age-related differences prevalence rates ofe-liquid exposures. For example, youth and young adults had a higher prevalence of e-liquid exposure than older adults. Further, higher e-cigarette use frequency was related to higher odds of e-liquid exposure, but this association was not observed in older age groups. Although increased e-cigarette use frequency may be related to frequent refills of e-liquids in both younger and older individauls, it is possible that younger individuals are less experienced than older adults in safely handling e-cigarette devices and liquids, . It is worth noting that although older adults had the lowest prevalence of e-liquid exposure tthey reported the highest prevalence of “becoming sick” and “going to a hospital” after the exposure This may suggest that the severity of e-liquid exposure might be worse among older adults than younger people, necessitating a greater level of medical attention. Alternatively, youth may experience similar types of e-liquid exposure, but be less likely to present to the hostipal due to confidentiality concerns to the hospitals and may hide their e-liquid exposure from parents and may go to the hospital for greater symptom severity. Future studies should use clinical data (e.g., electronic health records) to identify specific symptoms and severity of symptoms related to e-liquid exposure. 

Page 8, Our study also highlights sociodemographic disparities in e-liquid exposures. Among young adults, Non-Hispanic individuals and those with higher income levels had higher odds of e-liquid exposure to e-liquid. In general, more frequent use of e-cigarettes may be associated with increased risks of e-liquid exposure. Existing data indicate that e-cigarettes are more likely to be used by non-Hispanic White individuals, and those from higher income levels among young adults.2 Thus, certain education and communication related to harmful effects and risks of e-liquid exposure could be targeted to populations who are at greater risk for using e-cigarettes.

7. In the abstract, I suggest including clear and specific definitions of groups and variables in the abstract, e.g., specifying the age range of ‘youth’, ‘young adults’, and ‘older adults’, defining ‘becoming sick’.

Response: We defined the age groups. “Becoming sick” was not defined in the PATH Study. Thus, we have added this to the limitation. 

Revised text:

Page 2, We also examined associations between these outcomes and the device type used (refillable tank /mod system, replaceable prefilled cartridges, disposable/ other device type). E-liquid exposure was reported by 25% of youth (aged 12-17 years), 25% of young adults (aged 18-24 years), and 19% of older adults (aged≥ 25 years).

Page 9, Survey questions did not assess the specific route of exposure, the severity of the exposure or subsequent sickness, or if going to the hospital resulted in hospital admission or other medical interventions. Future studies should examine these factors since the health impact and harm could vary by route and extent of exposure.

8. The phrase “vs a disposable device and something else” needs to be clarified; I understand this is the reference group including two collapsed options but the current phrasing is confusing without context from methods. For instance, it could be phrased as: “vs people who used other device types including disposables” and combined in the same parentheses as the aOR/CI. 

Response: We revised the sentence.

Revised text:

Page 2, Among young adults, the use of a refillable tank /mod system was associated with higher odds of e-liquid exposure (aOR=2.2, 95% CI=1.2, 4.1) than the use of other device types, including disposables.

9. In the abstract the conclusion is a little narrow (warning labels only); consider other packaging regulations too, and even device/bottle designs which could avoid leaks (this can be explored in more details in the discussion, with the caveat that you don't know the exact cause of exposure).

Response: We revised the abstract accordingly. 

Revised text:

Page 2, The findings suggest that, at a minimum, e-cigarettes/e-liquids may need warning labels that state the risks of e-liquid exposure and packaging regulations that promote device and bottle designs that minimize e-liquid spills.

10. Introduction: I cannot find the number of cases cited in the linked source. It would also help to specify the timeframe this number refers to (e.g., 3,864 new cases over the past year?)

Response: We updated the number with most recent data.

Revised text:

Page 3, Between 2011 and 2024, 50,743 e-cigarette- and nicotine liquid-related exposure cases were reported to Poison Control Centers in the United States.4

11. I suggest not specifying past 12 months/ past 30 days in this paragraph, because the discrepancy between to two raises questions which are not answered until the methods section, and it is redundant if specified below.

Response: We deleted this part.

12. Last sentence: Clearly state the comparators here; ‘tank and mod devices allow individuals to easily refill the device’ as compared to which devices? Disposables, all other devices? 

Response: We deleted this sentence in the revision.

13. Providing some background on the types of devices available and their characteristics may strengthen the rationale for this hypothesis while also providing clarifications for those less familiar with the e-cig market.

Response: We added the rationale for the study.

Revised text:

Page 3, We also examined associations between the outcomes and the device type used (refillable tank /mod system, replaceable prefilled cartridges, disposable/other device type) to explore whether the ability to manually manipulate e-liquids through certain product features might affect the risk of exposure. Understanding e-liquid exposure and related outcomes by device types is important since individuals can access a variety of e-cigarette device types in the United States and e-cigarette popularity has been changed in recent years such that disposable and mod systems are frequently used by adolescents and young adults in recent years.5 Further, mod systems might put people at greater risk for e-liquid exposure since people are directly putting e-liquids in their devices. Although closed devices such as disposables/cartridges have lower likelihood of e-liquid exposure since e-liquid is prefilled, research shows that youth and young adults are manipulating or ‘hacking’ these devices to put their own e-liquids, which may increase their e-liquid exposure.3

14. In the introduction it states “We hypothesized that using refillable tank or mod system e-cigarette devices would be associated with a higher likelihood of e-liquid exposures, given that these devices allow individuals to easily refill the device with e-liquids”, then in the discussion “Consistent with our hypothesis, use of refillable tank/mod systems or replaceable prefilled cartridges (vs. disposable devices or something else) was associated with higher odds of e-liquid exposure among young adults”. 

I don’t see how prefilled cartridge models are ‘consistent’ in this case; they were not mentioned in the hypothesis, and they do not require directly handling e-liquids.

Response: We revised this sentence.

Revised text:

Page 8, We also found that the use of refillable tank/mod systems or replaceable prefilled cartridges (vs. disposable / other device type) was associated with higher odds of e-liquid exposure among young adults.

15. Paragraph 1: Please provide additional details about PATH methodology or add a reference to a paper which does. All data used in this study was self-reported via surveys this should be specified in the opening paragraph.

Response: We added more details about the PATH Study.

Revised text:

Page 4, We conducted a secondary data analysis of the US nationally representative Population Assessment of Tobacco and Health (PATH) Study Wave 5 (2018-2019) youth (aged 12-17 years) and adult (aged 18 years and older) datasets. The PATH Study public dataset’s self-report measures are well-validated, and the details of the study methodology, eligibility criteria, measures, and theoretical framework are described elsewhere.6,7 The PATH uses multi-stage sampling to allow the calculation of US population estimates.

16. Please specify which statistical software/language was used.

Response: We specified the statistical program was used.

Revised text:

Page 6, We used STATA 18.0 (College Station, TX) for statistical analyses.

17. ‘becoming sick with devices with…’: rephrase e.g., ‘Older adults who used devices with refillable tanks, mod systems or replaceable prefilled cartridges had lower odds of becoming sick’

Response: We revised this sentence.

Revised text:

Page 7, Among older adults who were exposed to e-liquid, use of refillable tank/mod systems (aOR=0.1, 95% CI=0.0, 0.5) or replaceable prefilled cartridge devices (aOR=0.1, 95% CI=0.0, 0.5) (vs. disposable/other device type) was associated with lower odds of becoming sick after an e-liquid exposure.

18. The limitations should mention how broad ‘becoming sick’ is as a measure

Response: We already had this limitation but elaborate them in the revision.

Revised text:

Page 9, PATH survey data may be prone to self-report, recall, or social desirability bias. Survey questions did not assess the specific route of exposure, the severity of the exposure or subsequent sickness, or if going to the hospital resulted in hospital admission or other medical interventions.

19. “Did not assess .. whether it required hospital admission” – slightly confusing phrasing since it did assess if people went to hospital. Perhaps rephrase to clarify it didn’t assess if these hospitalizations were necessary?

Response: We revised the sentence.

Revised text:

Page 9, Survey questions did not assess the specific route of exposure, the severity of the exposure or subsequent sickness, or if going to the hospital resulted in hospital admission or other medical interventions. Future studies should examine these factors since the health impact and harm could vary by route and extent of exposure.

20. Could cite https://pubmed.ncbi.nlm.nih.gov/35622007/

Response: We cited this paper in the introduction.

Revised text:

Page 3, We also examined associations between the outcomes and the device type used (refillable tank /mod system, replaceable prefilled cartridges, disposable/other device type) to explore whether the ability to manually manipulate e-liquids through certain product features might affect the risk of exposure. Understanding e-liquid exposure and related outcomes by device types is important since individuals can access a variety of e-cigarette device types in the United States and e-cigarette popularity has been changed in recent years such that disposable and mod systems are frequently used by adolescents and young adults in recent years.5 Further, mod systems might put people at greater risk for e-liquid exposure since people are directly putting e-liquids in their devices. Although closed devices such as disposables/cartridges have lower likelihood of e-liquid exposure since e-liquid is prefilled, research shows that youth and young adults are manipulating or ‘hacking’ these devices to put their own e-liquids, which may increase their e-liquid exposure.3

21. “especially directed at specific device types”: would be good to elaborate on this point.

Response: We revised the sentence.

Revised text:

Page 9, This study provides one of the first estimates of e-liquid exposure and potential health burdens from the exposure among youth and adults using US nationally representative samples. Overall, the findings highlight the need for effective e-liquid packaging and labeling regulations. This could include warning labels to inform potential risk of e-liquid exposure, e-cigarette device, and e-liquid bottle design to prevent accidental exposure to e-liquids. Our findings also suggest that e-liquid 

---

## [Editor Report · Decision Letter 3]

1 Jul 2024

PONE-D-23-26133R3Unintended exposure to e-liquid and subsequent health outcomes among US youth and adultsPLOS ONE

Dear Dr. Lee,

Thank you for submitting your manuscript to PLOS ONE. After careful consideration, we feel that it has merit but does not fully meet PLOS ONE’s publication criteria as it currently stands. Therefore, we invite you to submit a revised version of the manuscript that addresses the points raised during the review process.

**ACADEMIC EDITOR: **There is confusion in table 2. Please Clarify them in foot notes such as AoR, ( ) (<0.01, 14.4). It means you have to mention which is AOR, 95% CI, p value and meaning of <0.01. Readers are confused with term such as <. and reason for using it. Next, if author highlights few more information about table 2 information.==============================

We look forward to receiving your revised manuscript.

Kind regards,

Umesh Raj Aryal, PhD

Academic Editor

PLOS ONE

Journal Requirements:

Additional Editor Comments:

There is confusion in table 2. Please Clarify them in foot notes such as AoR, ( ) (<0.01, 14.4). It means you have to mention which is AOR, 95% CI, p value and meaning of <0.01. Readers are confused with term such as <. and reason for using it. Next, if author highlights few more information about table 2 information.

Thank you and good luck

---

## [Author Response · Author response to Decision Letter 3]

4 Jul 2024

PONE-D-23-26133R3: Unintended exposure to e-liquid and subsequent health outcomes among US youth and adults

Additional Editor Comments:

There is confusion in table 2. Please Clarify them in foot notes such as AoR, ( ) (<0.01, 14.4). It means you have to mention which is AOR, 95% CI, p value and meaning of <0.01. Readers are confused with term such as <. and reason for using it. 

Response: We revised the Table 2 accordingly.

Revised text: Please see updated Table 2.

Next, if author highlights few more information about table 2 information.

Response: We have included all significant findings in the Results and Discussion. We are unclear what the editor is requesting regarding the request for “more information about Table 2 information”. Regardless, we are happy to respond to the request if it is further clarified or expanded upon.

---

## [Editor Report · Decision Letter 4]

10 Jul 2024

Unintended exposure to e-liquid and subsequent health outcomes among US youth and adults

PONE-D-23-26133R4

Dear Dr. Lee,

We’re pleased to inform you that your manuscript has been judged scientifically suitable for publication and will be formally accepted for publication once it meets all outstanding technical requirements.

Kind regards,

Umesh Raj Aryal, PhD

Academic Editor

PLOS ONE

Additional Editor Comments (optional):

No  further comments
---

## [Editor Report · Acceptance letter]

20 Jul 2024

PONE-D-23-26133R4 

PLOS ONE

Dear Dr. Lee, 

I'm pleased to inform you that your manuscript has been deemed suitable for publication in PLOS ONE. Congratulations! Your manuscript is now being handed over to our production team.

Kind regards, 

on behalf of

Dr. Umesh Raj Aryal 

Academic Editor

PLOS ONE